# Climate Analogues for Temperate European Forests to Raise Silvicultural Evidence Using Twin Regions

**Tobias Mette [1,*], Susanne Brandl [1] and Christian Kölling [2]**

[1] LWF—Bavarian State Institute of Forestry, Hans-Carl-von-Carlowitz-Platz 1, 85354 Freising, Germany; susanne.brandl@lwf.bayern.de

[2] AELF Roth—Food, Agriculture and Forestry Office Roth, Johann-Strauß-Str. 1, 91154 Roth, Germany; christian.koelling@aelf-rh.bayern.de

[*] Correspondence: tobias.mette@lwf.bayern.de; Tel.: +49-8161-4591-223

**Abstract:** Climate analogues provide forestry practice with empirical evidence of how forests are managed in "twin" regions, i.e., regions where the current climate is comparable to the expected future climate at a site of interest. As the twin regions and their silvicultural evidence change with each climate scenario and model, we focus our investigation on how the uncertainty in future climate affects tree species prevalence. We calculate the future climate from 2000 to 2100 for three ensemble variants of the mild (representative concentration pathway (RCP) 4.5) and hard (RCP 8.5) climate scenarios. We determine climatic distances between the future climate of our site of interest 'Roth' and the current climate in Europe, generating maps with twin regions from 2000 to 2100. From forest inventories in these twin regions we trace how the prevalence of 23 major tree species changes. We realize that it is not the '*how*' but the '*how fast*' species' prevalence changes that differs between the scenario variants. We use this finding to develop a categorization of species groups that integrates the uncertainty in future climate. Twin regions provide further information on silvicultural practices, pest management, product chains etc.

**Keywords:** climate analogue; climate change; model ensemble; twin region; analogue region; national forest inventories; species suitability; forest adaptation; forestry practice; Europe

## 1. Introduction

Climate change is a major threat to European forests [1,2]. The mean annual temperature has increased 1.7–1.9 °C from the pre-industrial reference to the last decade—far stronger than the global average of 0.94–1.03 °C [3]. Climate models predict temperature to rise another 0.9–3.5 °C until 2100 (climate models in Table 1 for Europe). While for Northern Europe temperature will continue to rise drastically, for Southern Europe a further reduction of summer precipitation will probably have more severe consequences [1,4]. Forests react to climate change not gradually but rather suddenly in response to climatic extremes [5,6]. Only recently, a series of three exceptionally dry summers has led to a widespread forest dieback in Middle Europe [7,8]. The dieback affected not only boreal tree species Norway spruce and Scots pine but also temperate tree species like European beech and others [9–11]. Foresters, forest owners and society are startled—is this the beginning of the extirpation of our forests? How could it come that far? Where will it lead us?

Forest science has a long history of investigating tree species-climate relations and of consistently warning about adverse consequences of climate change [12–15]. There may still be two strong arguments why forest owners hesitate to adapt their forests to a new climate. First, climate change remains an abstract risk as long as people are not affected directly [16,17]. Second, the introduction of new tree species requires daring to do something unknown and local knowledge is scarce [18,19]. It is this second argument that our article is targeting because climate analogues help bridging this knowledge gap by

identifying regions where such new species are a silvicultural reality today—in a climate comparable to what we expect in future.

Climate analogues are well-established in many fields of climate impact research [20–39]. The most common way climate analogues are used today are spatial analogues [29]. Spatial analogues define a site of interest, identify its possible future climate, and search for regions with a similar, i.e., *analogue*, climate today. As in [31,37] we adopt the term "twin regions" for these analogue regions. While the term climate analogue was coined in the mid-twentieth century, the principle of climate analogues is much older. It becomes evident, for instance, in the knowledge that cultivation success of introduced plant species depends on the climatic similarity of source and target regions [40]: "foreign wood species should only be planted where the climate is most closely related to the original climate zone". The first explicit climatic analogues were applied as agro-climatic analogues to speed up "rehabilitation of wartime devastated agricultural areas" by considering the introduction of new plants [20]. Until today, climate analogues remained popular in agriculture [21,30]. In the late 1980s, with increasing awareness of climate change, scientists realized the potential of climate analogues for climate *change* impact research [22]. The analogue where the current climate "here" ($A_c$) is similar to the current climate "there" ($B_c$) receives a temporal component. Most prominent are analogues of the type $A_f - B_c$, future climate here ($A_f$)−current climate there ($B_c$). But also $A_c - B_f$ or $A_f - B_f$ pairs can be formed [30]. The $A_f - B_c$ pair is also the one applied in climate change impact research in forestry [23,28,33–35,38]. Whereas foresters use analogues to focus on tree species, provenances, growth and management, the ecologists think in terms of species communities, no-analog species reassemblies and habitat shifts [25–27]. Also, city analogues have become popular both as a mechanism of awareness raising and a tool for developing adaptive strategies [24,32,37,39].

Despite the different aims, technically, all of the aforementioned studies rely on temperature and precipitation—ranging from simple annual aggregations as in [27], to more complex aggregations like interannual variability and parameter sets as in [37]. All studies employ some kind of similarity or dissimilarity definition—ranging from a simple binning [27] to normalized indices like a standardized Euclidean distance [30,32,41]. Whether simple or complex, what counts is that: "The appropriateness of a specific analogy in a specific situation [ . . . ] does not concern the number of similarities two objects share but rather the significance of the similarities". [22]. For the future climate at the site of interest, most of the aforementioned studies work with climate ensembles and two or more scenarios to account for the uncertainty in future climate pathways [42]. Depending on the chosen climate scenarios, parameters, dissimilarity metrics and handling of topography some of the studies translate the climate shift into a geographical shift which lies in the order of 0.5–5 km per year [26,27,43,44]. References [43–47] warn that natural tree species migration is too slow fueling a controversial debate on assisted migration [48–53].

The project ANALOG employs climate analogues as a means of communication and guidance to forest owners to promote forest adaption to climate change [54]. The results of the project have been published in different formats and for different sites [55–58]. The project is being realized in the region of Nuremberg, Germany, in cooperation between forest owner organizations and the local forestry administration which is situated in the city of Roth. A brief characterization of the methods behind our climate analogues is important to understand the development of our research objective (details/critics in methods and discussion sections). In the project ANALOG, we derive climate analogues for temperate forest sites in Europe, and, also, the search radius for climatically analogue regions focuses on Europe. The climate parameters we chose for the analogues are summer temperature, winter temperature and summer precipitation. The climate scenarios cover the mild representative concentration pathway (RCP) 4.5 and hard RCP 8.5 from an ensemble of 10 climate models each RCP. What is unique about our approach is that we resolve the projected climate change into *climate trajectories* from 2000 to 2100—which, in their spatial analogues, translate into *geographic trajectories* of twin regions. A pan-European data set on tree species occurrence [59] allows us to determine species prevalence in those regions and

to thereby relate *species prevalence trajectories* from 2000 to 2100 to the climate trajectories. For the most abundant tree species we realized that these prevalence trajectories are astonishingly smooth—considering that they are not modelled but purely empirical.

More than the typical "start-to-end" analogues that compare a current "start" climate with some "end" climates at the end the 21st century, our trajectories emphasize the dynamic character of climate change and the transient character of any forest management decision. Adding to this the uncertainty of possible future climate scenarios and differences between climate models makes it really a "tough nut" for anyone who wants to convey a clear message to the forest owner how to adapt his/her forest to climate change. However, we noted earlier that the species prevalence trajectories from 2000 to 2100 of the RCP 4.5 ensemble coincide with the species prevalence trajectories of the RCP 8.5 from 2000 to approximately 2060 [56,57]. In other words, the prevalence patterns, i.e., which species prevalence declines or inclines, are very similar. It is rather the time axis of the prevalence patterns, i.e., how fast species prevalence declines or inclines, that differs between the RCP 4.5 and RCP 8.5. This may, of course, be pure coincidence due to the chosen ensemble mean, but is it also true for other variations of the model ensemble?

For this study, therefore, we expanded our ensemble means with a 'low' and a 'high' variant by assigning different weights to the underlying climate models. Our research objective was to (1) find out how this uncertainty in the climate future affects the prevalence of 23 major tree species, and (2) to draw conclusions on how to adapt forests to climate change in terms of species choice. The investigation is demonstrated for the site 'Roth' named after a city near Nuremberg, Germany, in the center of the focal region of project ANALOG (Figure 1).

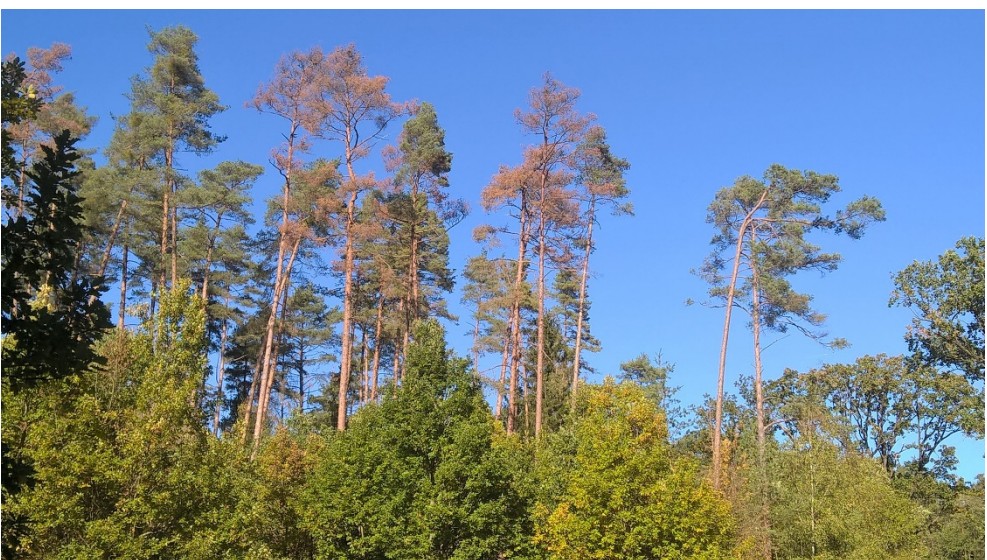

**Figure 1.** Mortal Scots pine (*Pinus sylvestris*) in the upper canopy, vital sessile oak (*Quercus petraea*) in the lower canopy. Forest impression from the study site 'Roth' near Nuremberg, Germany, following two exceptionally dry summers (Kölling, 2019, late summer).

## 2. Materials and Methods

Climate data are the backbone of climate analogues. To find climate analogues (or "twins") requires comparing current climate data and future climate data [22]. The current climate can be regionalized from measurement-based data sources with high accuracy. In contrast, uncertainty is part of the nature of the future climate projections [42]. Spatial, temporal and parametric resolution, extent and accuracy of the data must be chosen according to the aim of the study, data availability, and processing resources.

For our climate analogues we restrict ourselves to the two most prominent, available, and intensively studied climate parameters: air temperature (2 m aboveground) and precipitation. A monthly temporal resolution is sufficient, for the spatial resolution we

need 1 km or 30 arc raster data. The spatial extent should at least cover Europe. The temporal extent should preferably cover the last decades in the case of the current climate and the 21st century in the case of the future scenarios. For spatial consistency the current climate should consist of one single data set. For the future scenarios we need to consider a range of emission scenarios and climate models to account for the uncertainty of future climate.

Evaluating our criteria, our choice for the current climate data fell upon CHELSA [60]. The data are based on a downscaling of ERA interim climatic reanalysis to a resolution of 30 arc" world-wide from 1979 to 2013, monthly. In the case of temperature, statistical downscaling is used, in the case of precipitation, "orographic predictors including wind fields, valley exposition, and boundary layer height" [60] are incorporated. The monthly resolution of the data gives more flexibility in terms of choosing a reference period than periodically aggregated data like worldClim [61,62].

For the future climate projection we downloaded 10 RCP 4.5 and 10 RCP 8.5 climate models from the EURO-CORDEX (definition) domain [63,64]. RCP stands for representative concentration pathway (of greenhouse gases in the atmosphere). The numbers 4.5 and 8.5 quantify the theoretical change in the Earth's radiation balance (in Watt per $m^2$). There are two more standard RCPs: the numbers 2.6 and 6.0 identify them as one very low and one intermediate pathway between the RCP 4.5 and 8.5. How sensitive the Earth's climate reacts can be calculated by complex global climate models (GCMs) can only be estimated within a certain range. The uncertainty of climate future, therefore, has two components: (a) which pathway will be realized, and (b) which model matches the climate processes most correctly (for more information please refer to [65]). The EURO-CORDEX initiative provides climate change projections of Europe where regional climate models (RCMs) were used to downscale global climate models (GCMs). Table 1 specifies the four GCMs and the three RCMs that were used for the RCP 4.5 ensemble and the RCP 8.5 ensemble of our study. To avoid processing efforts for (bias) adjustment and self-analogue verification [66] we selected the adjusted model data in EUR11-resolution (~12 km) from the CLIPC project [67]. The adjustment was carried out based on distribution [68] and used EURO4M-MESAN data [69] from 1989–2010 as a reference. At the local scale, climate may still vary considerably due to topoclimatic effects. To account for this local climate variability we employ a simple delta adjustment method [70] that calculates the local difference in mean annual temperature and the ratio of the annual precipitation sum of the 12 km resolved climate model data to the 1 km resolved CHELSA data using 1989–2010 as common reference period.

**Table 1.** Climate models in this study. Global climate models (GCMs) and regional climate models (RCMs) used to downscale the GCMs (n = 10 representative concentration pathway (RCP) 4.5 models and 10 RCP 8.5 models). Data from EURO-CORDEX in EUR 11 resolution [63,64], monthly 1971–2100, adjusted as in [67].

| Global Climate Models | Regional Climate Models | | |
|---|---|---|---|
| | CCLM44-8-17, CLM-Community [71] | RCA4, Rossby Centre, Norway [72] | REMO, GERICS, Germany [73] |
| CNRM-CM5, CERFACS, France [74] | r1 (RCP 4.5/8.5) | r1 (RCP 4.5/8.5) | |
| EC-Earth, European consortium [75] | r12 (RCP 4.5/8.5) | r12 (RCP 4.5/8.5) | |
| HadGEM2-ES, Hadley Center, UK [76] | r1 (RCP 4.5/8.5) | r1 (RCP 4.5/8.5) | |
| MPI-ESM-LR, MPI-M, Germany [77] | r1 (RCP 4.5/8.5) | r1 (RCP 4.5/8.5) | r1 (RCP 4.5/8.5) r2 (RCP 4.5/8.5) |

We focus our climate analogues on three seasonally aggregated climate parameters, following standard meteorological definitions of summer (June to August) and winter (December to February):

- summer temperature: mean temperature from June to August;
- winter temperature: mean temperature from December to February;
- summer precipitation: precipitation sum from June to August.

These parameters are important for the forest distribution in temperate and boreal regions where temperature limits the vegetation period to 4–8 months [78] (pp. 41–44) and proved successful and robust in predicting tree species distribution and growth in temperate Europe [79,80].

For the current climate we average the aforementioned climate parameters from the CHELSA data from 1989 to 2010 (the adjustment period of the climate models). The current climate serves as a reference in two ways: (a) as the climate where we look for twin regions of the projected future climate of a site, (b) as the reference climate which we use for a local adjustment of the climate models.

Climate trajectories from 2000 to 2100 are calculated for each of the aforementioned climate parameters and for each of the 20 climate models (10 for RCP 4.5, 10 for RCP 8.5). Each trajectory consists of six 20-year time steps (average of 20-year intervals): 2000 (1991–2010), 2020 (2011–2030), . . . , 2100 (2091–2110). Since there are no data beyond 2100, we make the following assumptions for the last interval: in the case of temperature the trend from 2071–2100 continues rising linearly to 2110; in the case of precipitation the average for 2091–2100 holds true for the entire interval.

In previous studies, we aggregated the RCP 4.5 and RCP 8.5 climate trajectories of our climate models to one mean trajectory for each RCP [54–58]. In this study we added two different aggregations for each RCP and distinguish them as 'low' and 'high' variants from the 'mean' variant. While in the mean variant all climate models are averaged with even weights, in the low and high variant weights are shifted towards models with a lower or a higher summer temperature in 2100 (Table S1). The weights are assigned by fitting a beta distribution on an assumed Gaussian distribution of summer temperature in the models. For the low variant, the beta distribution is left-sided and its weights return the negative standard deviation, for the high variant vice versa. Looking at Table S1 one can see that the low variant weights the CNRM models strongest and the high variant the HadGEM models. The same model weights are also applied to winter temperature and summer precipitation. Effectively, the low, mean, and high variants contain different shares but all models. This procedure reduces decadal variabilities and conserves the covariation between the climate parameters.

Focal region of the project "ANALOG" is the region of Nuremberg where the climatic "hot" spots (by its actual meaning) are the Pegnitz and Rednitz valleys. This region forms a relatively homogeneous climate class in the Keuper hill-lands (upper Triassic) [58]. As a consequence of the sandstone bedrock the substrates are very nutrient-poor and sandy. Scots pine has a long tradition in that area and is encountered on 93% of the plots of the national forest inventory 2012 (n = 76, Figure 1) [81]. Spruce and pedunculate oak were surveyed on less than 30% of the plots, birch, beech and hornbeam on less than 10%. Overall forest cover is 46%. The coordinates for the demonstration site of this study, the city of 'Roth' are 49.246, 11.092 (lat/lon WGS84); altitude is 340 m a.s.l. As displayed in Table 2, the climate in 2000 (1991–2010) had an annual mean temperature of 9.5 °C and an annual precipitation sum of 677 mm. With 18.3 °C in summer and 0.9 °C in winter, and a precipitation maximum in summer (208 mm), the climate can be characterized as warm-humid (sub)continental. In 2100, the summer temperature has risen by +1.3 °C (RCP 4.5 low) to +5.5 °C (RCP 8.5 high) with almost equal intervals between the variants. Due to the covariation between the climate parameters, both in the case of RCP 4.5 and RCP 8.5, the models with a high rise in summer temperature are also the models with a stronger reduction in summer precipitation. In 2100, the summer precipitation has changed by +4% °C (RCP 4.5 low) to −22% (RCP 8.5 high). The delta in winter temperature is almost equal within the RCP variants (RCP 4.5 +2.4 °C, RCP 8.5 +4.9 °C). This has an important consequence: (a) in the high variants summer temperature rises stronger than

winter temperature: climate becomes more continental, and (b) in the low variants summer temperature rises less strongly than winter temperature: climate becomes more oceanic.

**Table 2.** Climate parameters 2000 and 2100 in the six RCP variants for the Roth site.

| RCP Variant | | Temperature (°C) | | | Precipitation (mm) | | |
|---|---|---|---|---|---|---|---|
| | Year | Annual | Summer | Winter | Annual | Summer | Winter |
| All | 2000 | 9.48 | 18.33 | 0.89 | 677 | 208 | 145 |
| RCP 4.5 low | 2100 | 11.20 | 19.67 | 3.32 | 789 | 216 | 176 |
| RCP 4.5 mean | 2100 | 11.55 | 20.35 | 3.25 | 750 | 213 | 172 |
| RCP 4.5 high | 2100 | 11.81 | 20.91 | 3.39 | 709 | 195 | 172 |
| RCP 8.5 low | 2100 | 13.08 | 21.49 | 5.6 | 836 | 209 | 223 |
| RCP 8.5 mean | 2100 | 13.85 | 22.85 | 5.85 | 797 | 190 | 214 |
| RCP 8.5 high | 2100 | 14.34 | 23.81 | 5.84 | 727 | 162 | 196 |

Figure 2a,b show the summer temperature trajectories and summer precipitation trajectories of the six RCP variants for the site Roth on a time axis from 1950–2100. Summer temperatures in the mean, low and high variants split up instantly after 2000, but the RCPs behind the variants only split up after 2040. Figure 2c plots the summer temperature and summer precipitation trajectories directly against each other.

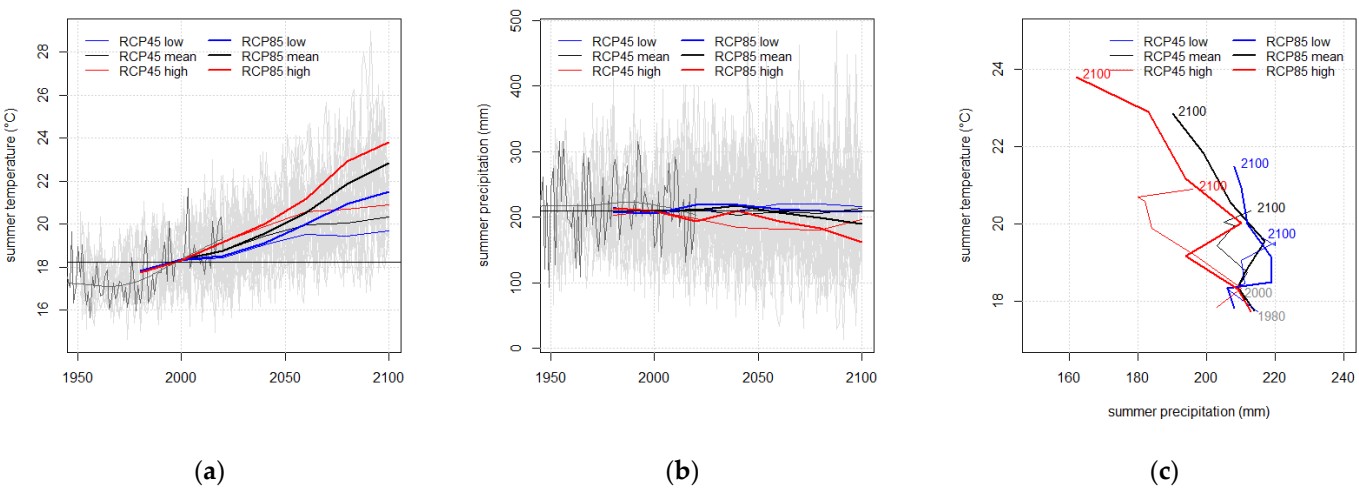

(**a**)                    (**b**)                    (**c**)

**Figure 2.** Climate trajectories for site Roth displaying the three RCP 4.5 and RCP 8.5 variants: (**a**) summer temperature 1950–2100, (**b**) summer precipitation 1950–2100, (**c**) summer temperature vs. summer precipitation. Noisy grey lines in (a/b) display the model data, the black line in (a/b) the gridded temperature and precipitation data from [82].

Despite some variation the relation is not random but follows an overall trend (or corridor) of decreasing summer precipitation with increasing summer temperature.

To quantify how similar two climates are, we need a climatic distance metric (dis-/similarity index). Plotting the three climate parameters summer temperature, winter temperature and summer precipitation in a Cartesian coordinate system a simple Euclidean distance can be calculated between any two points by extracting the square root of the sum of the squared differences for each parameter. This procedure is very common [41]. The question, however, is how to scale the parameters to each other—clearly, trees are much less sensitive to 1 mm difference in summer precipitation than 1 °C difference in summer temperature. A common approach is a normalization, e.g., by the standard variation over a 30-year time period [32]. To find a normalization that reflects the tree species' sensitivity to our three climate parameters we evaluated species distribution models of 33 tree species in Europe (Mette, unpublished). Employing strongly penalized generalized additive models [44] we can determine the tree species sensitivity over the range of each climate

parameter. For the climate parameter range in this study (Table 2) we can approximate the normalization as follows:

$$climDist \sim \sqrt{\text{diff}\left(t_{jja}/0.7\right)^2 + \text{diff}\left(t_{djf}/1.1\right)^2 + \text{diff}\left(p_{jja}/40\right)^2} \tag{1}$$

with *climDist* = climatic distance, $t_{jja}$ = summer temperature in °C, $t_{djf}$ = winter temperature in °C and $p_{jja}$ = summer precipitation in mm. In Equation (1), a climatic distance of 1 [unit] corresponds to 0.7 °C difference in summer temperature or 1.1 °C difference in winter temperature or 40 mm difference in summer precipitation or—as an example for a combination: 0.4 °C diff($t_{jja}$), 0.6 °C diff($t_{djf}$) and 25 mm diff($p_{jja}$). Note, that the climatic distance quantifies only the magnitude but not the direction of a climatic difference, i.e., contains no information which climate parameter deviates how strong and whether the deviation is positive or negative. As long as the relation between the normalization constants is respected, the quantity of the scaling parameters is not important. It would only change the magnitude of the climatic distance. For the constants in Equation (1), 90% of Germany lies within a climatic distance below 3.5 [units] from the reference climate of the site Roth (cf. Figure S1).

Twin regions are regions where the current climate is very similar to the future climate of a certain site of interest (for a selected set of climate parameters). In our case, we determined twin regions in the current climate in Europe (CHELSA 1989–2010) for each of the six 20-year time steps (2000–2100) of each of the six variants. We generate one twin regions maps for each RCP variant and assign the different time steps distinct colors. Regions that are analogue to two or more time steps are assigned to the latest one. We thereby transform the climate trajectories into spatial (geographic) trajectories of twin regions. The term "trajectory" is of limited adequacy as the twin regions do not tend to align in an ordered way but are rather dispersed due to the strong influence of topography. No climate is absolutely identical, and twin regions must be defined as regions within a certain climatic distance. The distance threshold is essentially a compromise. It should be loose enough to provide statistically robust information and strict enough to exclude as too distant, useless information. We set the threshold according to our requirements on a robust estimate of tree species prevalence to a value of 1.5 [units]. Once the twin regions are defined, they can be studied in more detail in terms of geology, soils, landscape etc. to find the most comprehensive possible match. Apart from scientific data exploitation, twin regions can be explored on site by everyone. The latter argument makes twin regions an extremely useful tool in communicating climate change and demonstrating ways to build climate-resilient future forests.

We are most interested in the tree species prevalence in the twin regions as an indication of which tree species are climate-resilient under the expected future climate of our site of interest. One of the most common approaches is climate-sensitive species distribution models (SDMs, [44,83–86]). For the climate analogues approach we can choose a more direct way by looking at the species spectrum in the twin regions. To estimate the tree species prevalence in the twin regions we used the national forest inventory data of 21 countries joined in a pan-European occurrence data set [59]. The data set is very handy as it is open access, harmonized in terms of species names and uses a common 1 km geographic reference grid. Nonetheless, differences in survey methods and grid densities of the national inventories may confound a comprehensive analysis [87]. To avoid such problems we first of all made sure that all the species in our analysis were actually part of the surveyed species spectrum in the national forest inventories (NFIs) of the twin regions. Second, we adjusted differences in grid density by applying a country-specific plot representation factor (km$^2$ forest area per plot, cf. Table S2)—well aware that regional grid differences cannot be reconstructed. What remains unsolved is that, for instance, larger plot sizes, clustered plots or smaller diameter thresholds all increase the probability for a species' occurrence—especially of rare species.

The tree occurrence data set yields a total number of 558,282 observations for 242 tree species in 21 countries. In our analysis we focus on 23 species that were among the three most abundant species in at least one time step of at least one RCP variant (Table 3). In the figures the species names were coded with an intuitive abbreviation of the scientific name (genus + species). For all 23 species we determine the absolute species prevalence $prev_{abs}$ from the occurrence data in the twin regions of each time step ($i$) and each RCP variant ($j$):

$$prev_{abs}(i,j) = \frac{\sum nPlots_{occ}(i,j) \times repFac}{\sum nPlots_{all}(i,j) \times repFac} \tag{2}$$

$nPlots_{occ}(i,j)$ refers to the number of plots in the twin region $(i,j)$ where the species occurs, $nPlots_{all}(i,j)$ to all plots, and $repFac$ to the NFI-plot representation factors that balances different densities between the countries. Once the absolute prevalence has been calculated for all time steps we can calculate a relative species prevalence $prev_{rel}$ from $prev_{abs}$ by dividing through the maximum prevalence of the species in any of the time steps of any RCP variant:

$$prev_{rel}(i,j) = prev_{rel}(i,j)/\max(prev_{abs}) \tag{3}$$

Like the favorability measure in SDMs, the relative species prevalence $prev_{rel}(i,j)$ ensures that each species maximum prevalence is set to 1 (=100%). It thereby normalizes different absolute prevalence between the species. Unlike in SDMs, the maximum prevalence $\max(prev_{abs})$ is derived only for the twin regions. The true maximum prevalence may lie outside the climate space covered by the twin regions. However, as long as we select the most abundant species we assume that each of the species is a valid silvicultural option and the trend of the relative occurrence reflects the climate-sensitivity correctly.

The relative prevalence of the most abundant tree species has become a standard output of the "ANALOG"-project [54–58]. In prevalence trajectory graphics we plot the relative prevalence of all species as horizontal bars (y-axis) from 2000 to 2100 (x-axis). The thickness of the bar indicates the relative prevalence at a certain time step. Decreasing thickness from left (2000) to right (2100) indicates decreasing prevalence in the twin regions along the climate trajectory, increasing thickness increasing prevalence. The species are ordered according to the year where they reach their maximum relative prevalence, separately for conifers and broadleaves. Grey numbers on the below the x-axis tell the number of plots in the twin regions for each 20-year time step. Grey numbers on the right vertical axis count the occurrences in all twin regions for each species (weighted by country-specific representation factor). Asterisks <*> in the species bars mark the three species with the highest absolute prevalence in each 20-year time step.

All analyses and graphics were done in R-Studio [88] with support of the raster and rgdal packages [89,90].

## 3. Results

### 3.1. Twin Regions Map

The twin regions maps in Figure 3 visualize how the future time-climate-trajectories for the site Roth turn into a spatial-geographic trajectory in Europe's current climate. Figure 3a shows the twin regions for the RCP 4.5 mean variant, Figure 3b for the RCP 8.5 mean variant. For the time steps 2000, 2020 and 2040 the RCP 4.5 and RCP 8.5 twin regions are very similar since their climate trajectories do not diverge much (c.f. Figure 2). The 2000 twin regions (grey) cover the vicinity of Roth itself—the Bavarian Triassic and Danube valley, large parts of the plains of Eastern Germany and Poland, and the Elbe valley in Czech Republic. The 2020 twin regions (light blue) are situated mainly in Eastern Germany and in the mountain ranges between the West-German Pfalz and the French Morvan. The 2040 twin regions (green) are concentrated in the Rhine-Main-valley and the lower altitudes bordering the French Morvan. A smaller 2040 twin region is formed by the Weinviertel in Austria and the bordering Moraval valley in Czech Moldavia. From 2040 onwards the RCP 4.5 and 8.5 analogues separate. In RCP 8.5, the upper Rhine valley remains mainly

a 2040 twin region (green), only the hottest spots become a 2060 twin region (yellow). In RCP 4.5, the upper Rhine valley covers twin region from 2040 (green) to 2100 (dark red). The Saone valley between Dijon and Lyon is a 2100 twin region in RCP 4.5, while in RCP 8.5 it is a twin region for 2040 at Dijon and 2060 at Lyon. Only the lowest parts of the rivers Saone and Rhone near Lyon are 2080 twin region (orange). 2100 twin regions in RCP 8.5 are found downstream the Rhone valley between Valence and Montelimar. Larger twin regions for 2080 and 2100 in RCP 8.5 can also be found in the upper Po valley in Italy and the Istrian peninsula in Croatia.

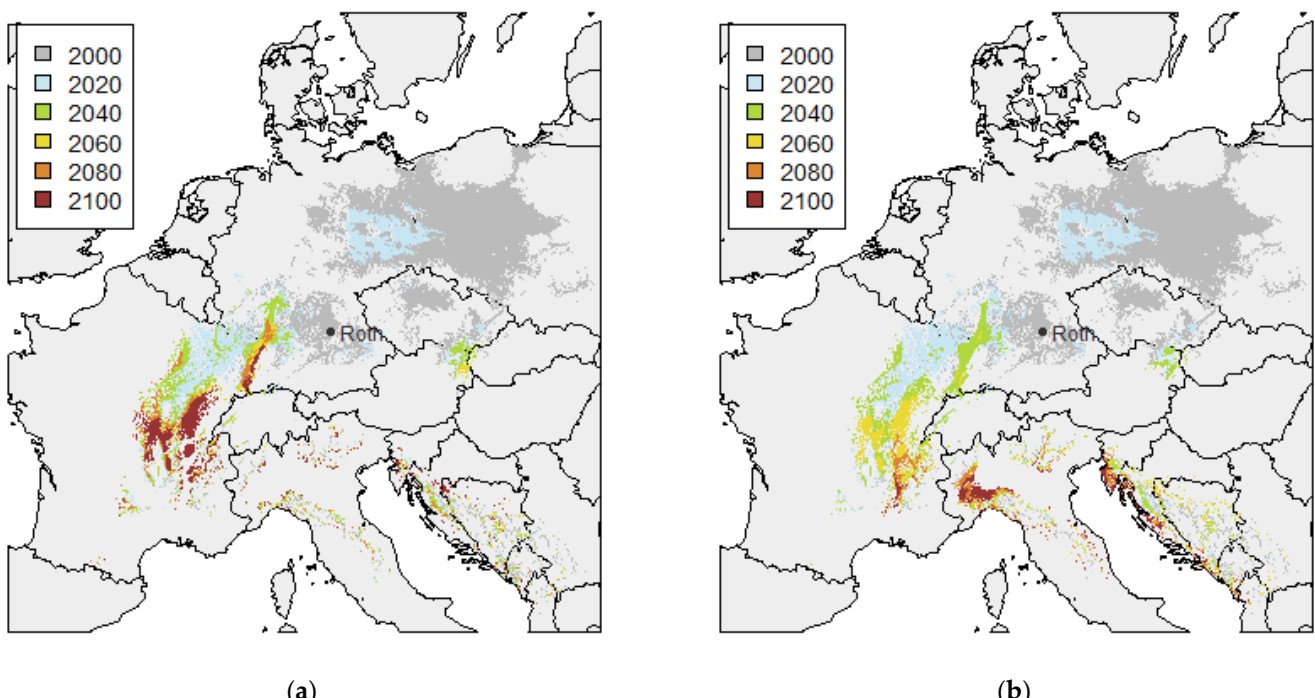

（**a**）　　　　　　　　　　　　　　　　　　　　　　（**b**）

**Figure 3.** Twin region maps for site Roth (**a**) RCP 4.5 mean variant, (**b**) RCP 8.5 mean variants.

The twin region maps in Figure 4 complement Figure 3 by visualizing the low and high variants of the RCP 4.5 and RCP 8.5. As pointed out in Table 2, the low, respectively, high variants exhibit a weaker, respectively, stronger increase in summer temperature 2100, and a weaker, respectively, stronger decrease in summer precipitation in 2100 than the mean variant.

Winter temperatures, by contrast, are very similar between the variants. This makes the low variants more oceanic and the high variants more continental compared to the mean variant. From a geographic perspective, the twin regions of the low variant should, therefore, shift northwards and of the high variant southwards compared to the mean variants. From a time perspective, the twin regions of the low variant should shift towards later times and of the high variant towards earlier times compared to the mean variant. In both the RCP 4.5 and the RCP 8.5 variants, the time shift is clearly visible: areas that are green (2040) or yellow (2060) in the low variant become blue (2020) or green (2040) in the high variant. The upper Rhine valley, for instance, is a 2060 twin region in the low RCP 8.5 variant but a 2040 twin region in the high variant. In the low RCP 4.5 variant, the valley is even dark red (2100) while green (2040) in the high variant. The geographic north-south shift, however, is hardly observable. More obvious is an east-west (continentality) shift between the variants. In the low RCP 8.5 variant, e.g., oceanic regions like the French Gascogne become twin regions. The twin regions in the lower Po-valley in the high RCP 8.5 variant are both the result of the higher summer temperature and continentality.

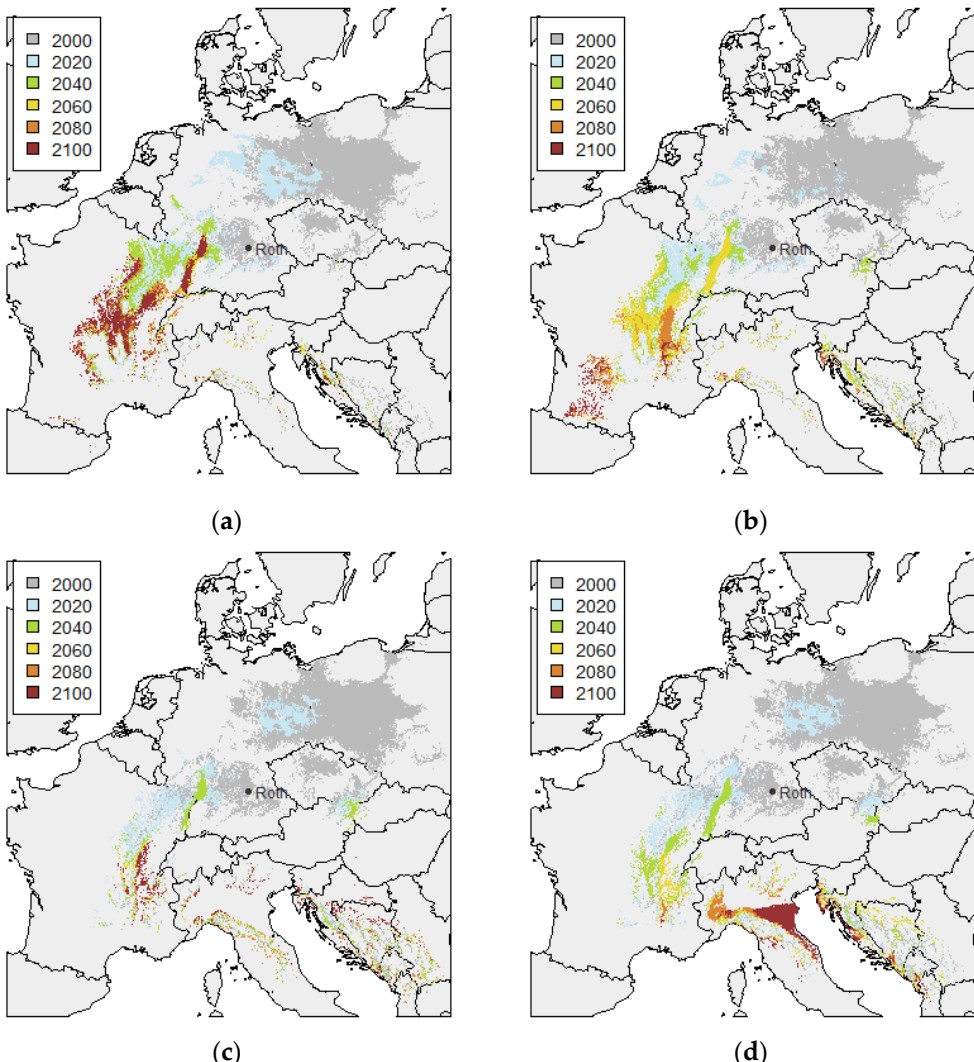

**Figure 4.** Twin region maps for site Roth (**a**) low RCP 4.5 variant, (**b**) low RCP 8.5 variant, (**c**) high RCP 4.5 variant, (**d**) high RCP 8.5 variant.

### 3.2. Prevalence Trajectory Graphics

The prevalence trajectory graphics in Figure 5 display the relative prevalence of 23 major tree species in the twin regions along the geographic trajectories of the RCP 4.5 and RCP 8.5 mean variant for the site Roth (corresponding to Figure 3). The graphics of the low and high RCP variants can be viewed in the supplement (Figure S2). The grey numbers below the axis indicate that the number of plots in the twin regions are highest for 2000 and 2020 and decrease towards 2100—in the RCP 8.5 much stronger than in the RCP 4.5. The grey numbers on the right vertical axis are high especially for species with high prevalence between 2000 and 2060. Low counts are typical for species with low prevalence in general or a high prevalence in the much scarcer 2080- and 2100-twin regions.

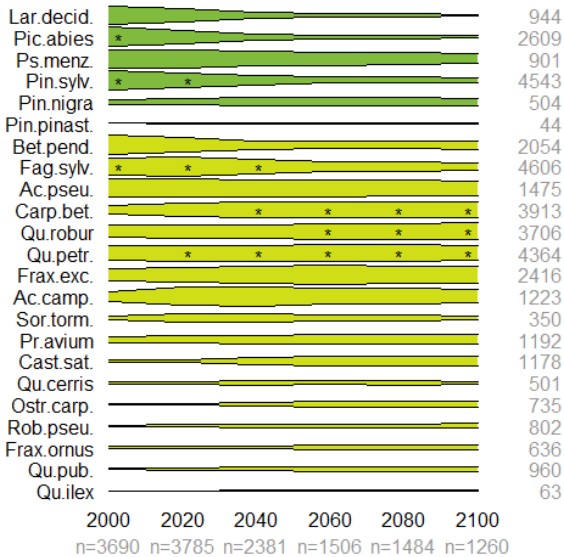

(**a**) Species prevalence trajectories for site Roth RCP 4.5 mean variant

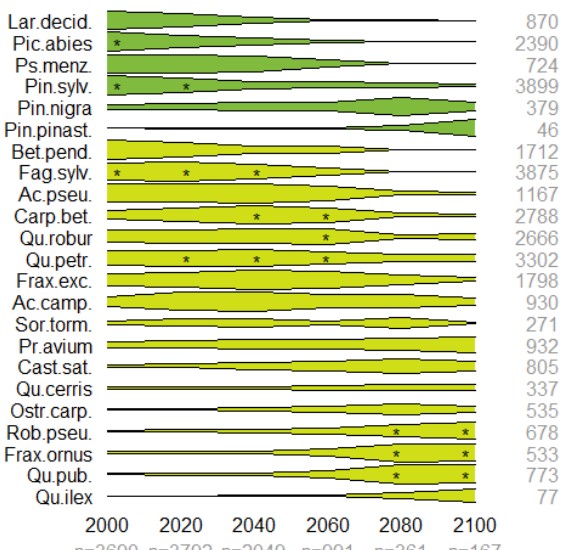

(**b**) Species prevalence trajectories for site Roth RCP 8.5 mean variant

**Figure 5.** Prevalence trajectory graphics showing relative prevalence from 2000 to 2100 of 23 major tree species in the twin regions of the RCP 4.5 and RCP 8.5 mean variants for the Roth site. Grey numbers below the x-axis indicate the number of national forest inventory-plots in the twin regions of each time step; grey numbers on the right vertical axis the total number of species occurrence in the plots of all twin regions from 2000 to 2100. Asterisks <*> in the species cones mark the three species with the highest absolute prevalence in each 20-year time step. Species abbreviations as in Table 3.

**Table 3.** Names and abbreviation of 23 major European tree species in the focus of our study. Prevalence data for selected countries from [59]. Prevalence = plot number with species/total plot number (%). FIN = Finland, NOR = Norway, SWE = Sweden, DEU = Germany, AUS = Austria, CHE = Switzerland, FRA = France, ITA = Italy, ESP = Spain.

| Species Name | | | Prevalence in Selected Countries' NFIs (%) | | | | | | | | |
| English | Scientific | Abbreviation | FIN | NOR | SWE | DEU | AUS | CHE | FRA | ITA | ESP |
|---|---|---|---|---|---|---|---|---|---|---|---|
| European larch | *Larix decidua* | Lar.decid. | 0 | 0 | 0.2 | 12.0 | 34.9 | 0 | 1.0 | 9.5 | 0 |
| Norway spruce | *Picea abies* | Pic.abies | 65.6 | 55.2 | 75.5 | 56.4 | 84.5 | 67.4 | 8.2 | 13.7 | 0 |
| Douglas fir | *Pseudotsuga menziesii* | Ps.menz. | 0 | 0 | 0 | 9.4 | 0.4 | 0.7 | 3.9 | 0.8 | 0.2 |
| Scots pine | *Pinus sylvestris* | Pin.sylv. | 84.3 | 47.2 | 75.9 | 39.1 | 24.3 | 8.8 | 12.6 | 6.3 | 14.3 |
| Black pine | *Pinus nigra* | Pin.nigra | 0 | 0 | 0 | 0.6 | 1.8 | 0.1 | 3.9 | 5.9 | 10.8 |
| maritime pine | *Pinus pinaster* | Pin.pinast. | 0 | 0 | 0 | 0 | 0 | 0 | 7.0 | 2.1 | 16.1 |
| common birch | *Betula pendula* | Bet.pend. | 27.4 | 3.5 | 5.7 | 23.0 | 9.5 | 5.3 | 9.3 | 2.8 | 0.2 |
| European beech | *Fagus sylvatica* | Fag.sylv. | 0 | 0.2 | 1.5 | 50.9 | 34.4 | 43.3 | 21.1 | 18.9 | 5.9 |
| mountain maple | *Acer pseudoplatanus* | Ac.pseu. | 0 | 0.1 | 0 | 16.0 | 14.7 | 19.1 | 4.5 | 6.9 | 0.2 |
| hornbeam | *Carpinus betulus* | Carp.bet. | 0 | 0 | 0.3 | 13.6 | 7.7 | 2.1 | 17.3 | 4.4 | 0 |
| pedunculate oak | *Quercus robur* | Qu.robur | 0.1 | 0 | 5.3 | 24.9 | 8.2 | 3.4 | 26.0 | 1.8 | 5.4 |
| sessile oak | *Quercus petraea* | Qu.petr. | 0 | 0 | 0 | 21.1 | 7.4 | 4.5 | 20.6 | 4.6 | 2.0 |
| Common ash | *Fraxinus excelsior* | Frax.exc. | 0.1 | 1.0 | 1.2 | 16.8 | 14.8 | 16.0 | 12.1 | 6.0 | 0.9 |
| field maple | *Acer campestre* | Ac.camp. | 0 | 0 | 0 | 2.3 | 1.7 | 1.3 | 6.3 | 7.7 | 0.8 |
| wild service tree | *Sorbus torminalis* | Sor.torm. | 0 | 0 | 0 | 0.6 | 0.3 | 0.1 | 2.9 | 1.4 | 0.1 |
| sweet cherry | *Prunus avium* | Pr.avium | 0 | 0.1 | 0.3 | 5.0 | 3.0 | 3.2 | 6.1 | 7.9 | 0.2 |
| chestnut | *Castanea sativa* | Cast.sat. | 0 | 0 | 0 | 0.8 | 1.4 | 3.3 | 11.9 | 16.3 | 3.8 |
| Turkey oak | *Quercus cerris* | Qu.cerris | 0 | 0 | 0 | 0 | 1.8 | 0.1 | 0.2 | 21.2 | 0 |
| hop hornbeam | *Ostrya carpinifolia* | Ostr.carp. | 0 | 0 | 0 | 0 | 0 | 0.5 | 0.2 | 20 | 0 |
| black locust | *Robinia pseudoacacia* | Rob.pseu. | 0 | 0 | 0 | 1.5 | 1.5 | 0.5 | 3.2 | 6.6 | 0.3 |
| Manna ash | *Fraxinus ornus* | Frax.ornus | 0 | 0 | 0 | 0 | 0 | 0.1 | 0.2 | 21.5 | 0 |
| pubescent oak | *Quercus pubescens* | Qu.pub. | 0 | 0 | 0 | 0 | 0.1 | 0.8 | 12.7 | 30.2 | 2.8 |
| holm oak | *Quercus ilex* | Qu.ilex | 0 | 0 | 0 | 0 | 0 | 0 | 5.1 | 10.1 | 26.1 |

From top to bottom the species list in Figure 5 starts with European larch and Norway spruce—both species with an early steep decrease even in the RCP 4.5 scenario although spruce is in 2000 still among the most abundant species. Douglas fir decreases already slower and maintains almost 50% of its highest prevalence in the RCP 4.5 2100. It declines sharply in the RCP 8.5 between 2060 and 2080. Scots pine is a special case. Although it appears to decline in a similarly steep way as larch and spruce, it stands out through its high absolute prevalence. It is the most abundant species in RCP 4.5 2000 and 2020 and keeps throughout all scenarios a higher absolute prevalence than Douglas fir (cf. Figure 6). Even in RCP 8.5 2100 it is present on 4% of all plots (compared to 52% in 2000). In contrast to Scots pine black pine increases from 2000 to 2100, especially in the more continental high variants of RCP 4.5 and 8.5 (Figure S2). Maritime pine only becomes prevalent in the RCP 8.5 2100 (2080 in the high RCP 8.5 variant). Among the broadleaved tree species silver birch is the first to decrease followed by European beech. Both decline in RCP 8.5 between 2060 and 2080. However, until 2040 beech is among the most abundant species in RCP 4.5 and 8.5. Mountain maple which is known for its preference of moist nutrient-rich sites decreases later than beech after 2080 (RCP 8.5). Hornbeam, pedunculate and sessile oak are the most abundant species between 2040 and 2100 in RCP 4.5 and between 2040 and 2060 in RCP 8.5 (pedunculate oak in the low RCP 8.5 variant even until 2100, cf. Figure S2). Common ash, field maple, wild service tree and wild cherry have in common the fact that they occur at all times in all scenarios. Common ash is the most prevalent of them except for the RCP 8.5 2100 where wild cherry becomes more prevalent. Chestnut, Turkey oak and European hop-hornbeam all exhibit prevalence values below 5% in 2000. Chestnut already reaches above 10% in 2040 and ranges among the most abundant species between 2060 and 2100 (high RCP 4.5 variant), 2080 and 2100 (low RCP 8.5 variant) and 2060 (high RCP 8.5 variant). Hop-hornbeam and Turkey oak play little role in the mean and low variants, but in the continental high variants they maintain a prevalence above 10% from 2040 onwards. Hop-hornbeam becomes one of the most abundant species from 2060 to 2100 in the high RCP 4.5 variant. Black locust, manna-ash and pubescent oak join the game only after 2060 in the RCP 8.5 (but 2040 in the high RCP 4.5 and RCP 8.5 variants). All three rank among the most abundant species in the mean and high RCP 8.5 variant in 2080 and 2100 (pubescent oak also in other variants). Field elm and holm oak start to increase as late as 2100 in the mean and high RCP 8.5 variant, but even here do not reach the prevalence of the three previous species.

### 3.3. Tree Species Absolute Prevalence

The prevalence trajectory graphics in Figures 4 and 5 show that for a given point in time the species spectrum clearly differs between the RCP variants. We now scrutinize how strong the differences are for a given point in climate, i.e., we substitute time on the x-axis by the climate parameter summer temperature. In Figure 6, this is done with the absolute prevalence of the 23 tree species using the data of the RCP 8.5 mean variant. The time scales below the x-axis relate the summer temperature in each of the six RCP variants to the summer temperature on the x-axis: The low RCP 4.5 variant exhibits the weakest temperature rise from 18.3 to 19.6 °C, and the high RCP 8.5 variant exhibits the strongest from 18.3 to 23.8 °C. The other RCP variants fill the gap almost evenly. Consequently, for the low RCP 4.5 variant we expect the weakest and slowest change in the species spectrum between 2000 and 2100, for the high RCP 8.5 variant the highest and fastest change. From a common start in 2000 (18.3 °C) with the dominance of Scots pine, European beech and Norway spruce, in 2100 the low RCP 4.5 variant has reached 19.7 °C and passes through the hornbeam, pedunculate and sessile oak optimum. Beech is still strong but already declining. In the RCP 4.5 mean variant 2100 (20.3 °C), hornbeam, pedunculate and sessile oak are still prevalent but decline, while pubescent oak and chestnut are on the rise and already exhibit high prevalence. Also, manna-ash, black locust and hop-hornbeam gain in prevalence. This trend continues through the high RCP 4.5 variant (20.9 °C, 2100) and the low RCP 8.5 variant (21.4 °C, 2100) with chestnut and hop-hornbeam reaching their

prevalence maximum in the low RCP 8.5 variant in 2100. In the RCP 8.5 mean variant, the species spectrum in 2100 (22.8 °C) is dominated by pubescent oak, black locust, manna-ash, field elm, wild cherry, holm oak, chestnut and hop-hornbeam.

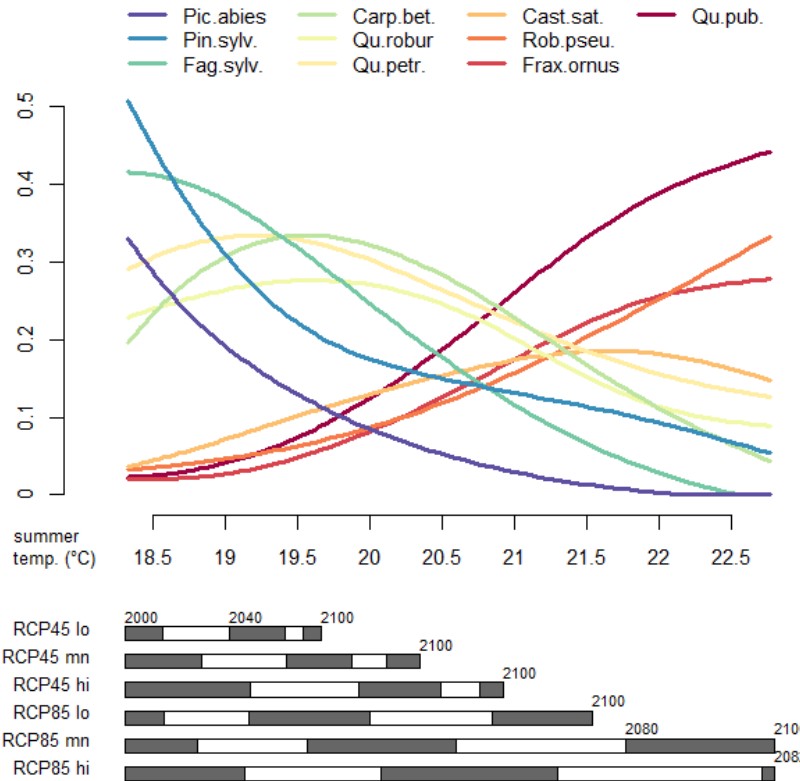

**Figure 6.** Absolute prevalence of 10 major tree species in the twin regions of the RCP 8.5 mean variant for site Roth, now with summer temperature on the x-axis. Time scales below the x-axis relate the summer temperature in the six RCP variants to the scale of the x-axis. Prevalence curves smoothed and interpolated using a smooth spline with 4 degrees of freedom.

Because the species prevalence plotted in Figure 6 corresponds to the RCP 8.5 mean variant it is important to know where there are differences in the actual species prevalence in the other RCP variants. This is done in Figure 7 for 12 of the 23 species with the same axes as in Figure 6. The thick black line represents the RCP 8.5 mean variant that was displayed in Figure 6. The general trend whether species' prevalence rises or declines is fairly similar between the variants. In particular, the prevalence of Norway spruce, Scots pine, European beech, chestnut, black locust and pubescent oak are close to the RCP 8.5 mean variant (>70% explained variance). Nonetheless, there are also interesting differences between the RCP variants. Hornbeam, pedunculate and sessile oak have higher prevalence in the low RCP variants (blue), and lower prevalence in the high RCP variants (red)—Turkey oak, hop-hornbeam, and manna-ash the opposite. Differences between the two variants are that (for a given summer temperature) the low variants have higher summer precipitation and winter temperatures (see Table 2 and Figure 2). The summer precipitation of the high RCP 4.5 variant in particular is between 2040 and 2080 lower than the RCP 8.5 mean variant. This leads to a temporary early and strong decrease in the case of hornbeam, pedunculate and sessile oak, and a temporary early and strong increase in Turkey oak, hop-hornbeam, manna-ash and pubescent oak in the high RCP 4.5 variant.

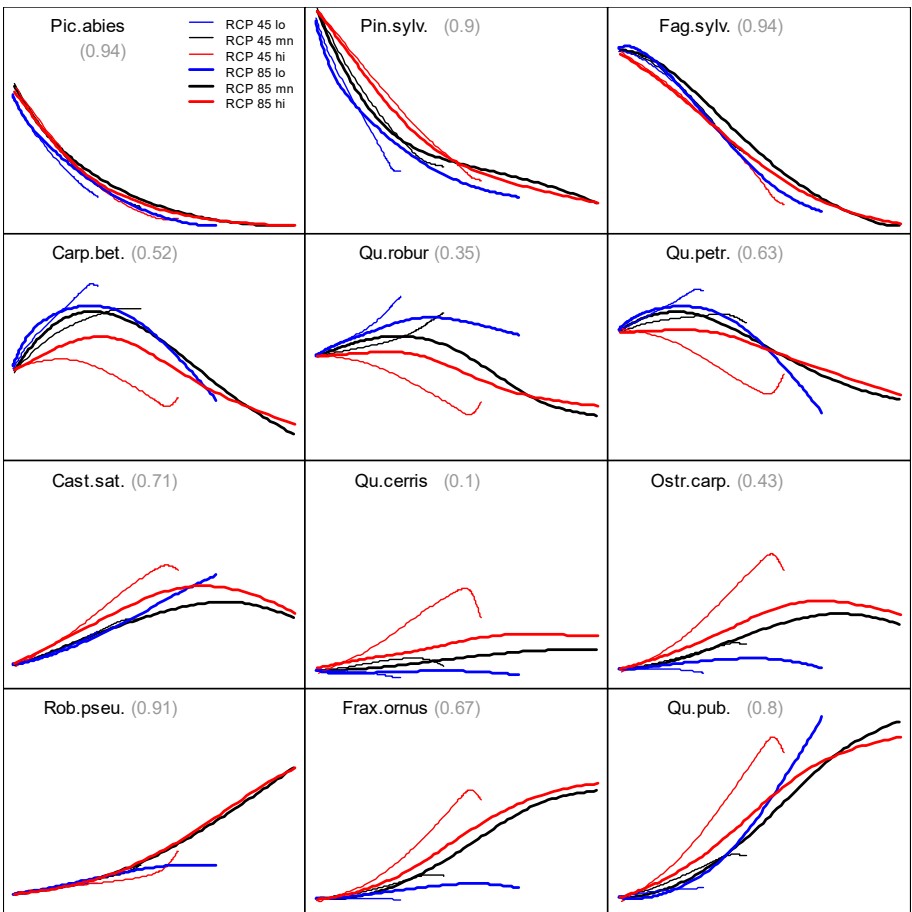

**Figure 7.** Absolute prevalence of 12 major tree species in the twin regions of the six RCP variants for the Roth site. The summer temperature on the x-axis scales from 18.3 to 22.8 °C, the absolute prevalence on y-axis from 0 to 0.5 (similar to Figure 6). Species abbreviations c.f. Table 3. Grey number in brackets behind species name indicates variance explained by RCP 8.5 mean variant. Prevalence curves smoothed and interpolated using a smooth spline with 4 degrees of freedom.

## 4. Discussion

In a brief retrospective, the results have presented how we use climate analogues to raise empirical evidence on species prevalence from "twin" regions. We did this by translating climate trajectories of different ensemble variants of the mild (RCP 4.5) and hard (RCP 8.5) climate scenario into geographic trajectories, and by translating these into species prevalence trajectories by raising species occurrence data in the twin regions. We will now discuss the results in terms of our research objectives: (1) how does the uncertainty in the climate future affect the prevalence of the 23 major tree species considered, and (2) what conclusions can be drawn to adapt forests to climate change in terms of species choice.

Our main result is that over a range of RCP 4.5 and RCP 8.5 ensemble variants—with temperature rises of 1.75 to 4.9 °C between 2000 and 2100—the prevalence curves of important tree species match very well when the time axis for each ensemble is scaled to a temperature axis (Figure 6). The prevalence curves resemble typical Gaussian-like dose-response functions [91], and start with a beech peak which turns into sessile oak-pedunculate oak-hornbeam peak which is followed by a strong rise in pubescent oak, manna-ash and black locust. Chestnut and hop hornbeam never dominate but have a prevalence peak between the temperate sessile/pedunculate and the Mediterranean pubescent oak.

In other words, the uncertainty in the climate future—as depicted by range of the scenario variants—does not affect so much the '*how*' but rather the '*how fast*' species' prevalence changes. There is a reason to it. The trajectories of the regarded climate

parameters summer temperature, winter temperature and summer precipitation lie within a more or less narrow "corridor" in all variants: with an increase in summer temperature, winter temperature also increases and summer precipitation decreases (see Figure 2c). Effectively, the projected future climate of site Roth becomes more Mediterranean (hotter and drier summers). This trend corresponds to the overall climate gradient from temperate in southern Europe—an advantage when searching climate analogues. Differences between the variants are mainly due to a different warming of summer and winter. The low RCP 4.5 and RCP 8.5 variants are more oceanic, the high variants more continental; the mean variants lie in between. Species like sessile oak, pedunculate oak and hornbeam have a more oceanic distribution at their southern distribution edge, species like Turkey oak, hop-hornbeam and manna-ash have a more continental distribution. The most notable deviation from the common climate corridor is the high variant of the RCP 4.5. From 2040 to 2080, summer precipitation is lower than the other variants. Due to the overall gradient of decreasing summer precipitation towards southern Europe the "twin" regions shift south—which has the same effect as an increase in temperature. The species prevalence curves in Figure 7 for the high RCP 4.5 variant are, therefore, advanced compared to the other variants.

The second objective challenges us in how these findings can help us in adapting forests to climate change. A first intuitive question may ask "what" this species spectrum should look like in 2100, 80 years from now. An optimist may take the RCP 4.5 as basis for his decision, a pessimist the RCP 8.5—who will be right? Well, it is the question that is wrong. More than anything the prevalence trajectory graphics (Figure 5) show that climate change and the species prevalence in the twin regions is dynamic and will also not stop in 2100. But if the difference between an RCP 4.5 and RCP 8.5 lies rather in the velocity (the "how fast") of the change (Figure 6) then the critical question for forestry is not "what" climate change brings but rather "how fast". Arguably, this is a very reductionist view, and we will certainly come back to the limitations. However, for now let us follow the line of thought because it has strong implications for climate adaptive forestry. We call it here—for the first time—the 0-3-0 principle based on a concept developed in [54–58].

For this 0-3-0 principle we divide the species spectrum of the prevalence trajectory graphics in Figure 5 into three categories. We assume the consensus that no matter what scenario we look at some species will be depreciated. In the case of the Roth site, larch and spruce prevalence strongly declines already in 2000 while maritime pine and holm oak prevalence starts to rise only beyond 2100 (RCP 4.5) resp. 2060 (RCP 8.5). These two edges of the species spectrum constitute the bordering 0s of the 0-3-0 principle. Scots pine may not necessarily count to the first 0 but requires at least caution; the decline in prevalence is strong and it is not clear how far the persistence even in climates of extremely hot and dry summers is due to local provenances [92–94]. The same doubts apply to birch, a typical pioneer species that is, however, in contrast to Scots pine, rather tolerated by forestry than actively promoted [95]. The focus of a climate adaptive forestry lies therefore on the species in the center of the prevalence trajectory graphics. These species can be further divided into three groups A, B, C—hence, the 3 in the 0-3-0 principle: (A) species with maximum prevalence today and 2020 (beech, mountain maple, Douglas fir), (B) species with a maximum prevalence until 2100 in RCP 4.5 or 2060 in RCP 8.5 (hornbeam, sessile oak, pedunculated oak, common ash, field maple, wild service tree, sweet cherry), (C) species with a maximum prevalence in 2080 or later in RCP 8.5, but rising after 2020 (chestnut, Turkey oak, hop hornbeam, black locust, manna ash, pubescent oak). Species from (A) are strong and vital today. Especially beech is so competitive that it will outgrow oak (B). However, even in RCP 4.5 it is (B) that has the best prognosis until the end of the century. Depending on their shade tolerance species from this group must be actively relieved from the competitive beech [96–98]. "Alternative" species from group (C) are typically not under cultivation today but in case of a stronger climate change may play a crucial role already towards the end of the century [99]. Therefore, we recommend enriching forests already today with these species alternatives. For these species, attention should

be paid not to selecting sites prone to late frost as arctic cold spells are common in Middle Europe [100,101]. The 0-3-0 principle has been promoted in media and communication of the ANALOG project under the terms "risky–secure–future" [56,58].

The categorization of tree species according to the 0-3-0 principle was derived from climate analogues. This analogue approach has some advantages and disadvantages, but the results are in accordance with other distribution-based approaches, the most prominent ones being species distribution models [44,83–86]. Species distribution models (SDMs) work with prevalence, too. In contrast to the analogue approach they define each species' own climate niche with a set of "personalized" variables—in some cases also including soil variables [44]. The prevalence in SDM is derived from integration over the entire species occurrence and not limited to a more or less narrow corridor as in analogues. This allows robust estimates also for less-abundant species. Therefore, if SDMs are more specific and stable, what is the benefit of analogues? We see the main benefit in the directness of the evidence. The explicit geographic realization in form of a climate twin makes it possible to see and visit the evidence [16,18]. To overcome century-old traditions requires daring something new—practical knowledge from twin regions can be critical for a success especially with the alternative species. This practical knowledge comprises information on species regeneration, growth, thinning, harvest, provenances, mixture, soil preferences, calamities, biodiversity, wood value chain etc. The absolute prevalence in Figure 6 also presents the silvicultural reality in the twin regions better than the relative prevalence. Ultimately, we recommend using analogue climates and SDMs as a source for mutual verification and complementary information.

What must be kept in mind, though, is that this silvicultural reality in the twin regions is also changing. For the Roth site, the upper Rhine valley was presented as a twin region for 2040–2060 in the RCP 8.5 mean variant. However, that was with respect to the climate conditions in the reference period 1991–2010. Twenty years later, the conditions have changed, especially as a consequence of the extremely dry summers of 2018–2020. The upper Rhine valley today would not present us with the 2040–2060 analogy but rather the 2060–2080 analogy in a period of strong change. Due to the adaption lag, forestry practices 20 years ago are still present in the legacy of the forests and foresters. However, the recent development has shown clearly the dynamics of forests under climate change. These dynamics contradict any search for a new final equilibrium in the next 150 years—even if we constrain greenhouse gas emissions to the rather mild RCP 4.5 scenario. Experience has taught us that we are not to expect gradual dynamics but rather sudden changes in reaction to climatic extremes [5,6]. Mixing species of different climatic niches like in the 0-3-0 principle reduces the risk of large-scale forest dieback and still permits a flexible adjustment towards a milder or harder climate change in the course of the 21st century.

## 5. Conclusions

Climate analogues for a wide range of future climate scenarios have been used to develop a practice-oriented categorization of species groups that integrates the uncertainty in future climate. We termed this concept the 0-3-0 principle, i.e., species with increasing risk (first 0), relatively secure species in medium terms (with 3 groups A, B, C) and species with increasing potential in future (second 0). Therefore, in the end, do we respond to the broad range of possible climate future scenarios with one single concept? Yes, we do, because there is no alternative. On the one hand, we cannot foresee the future and any if-then options like "if climate change comes mild then . . . , but if climate change comes hard then . . . " are basically another question and not an answer. On the other hand, to exclude species of (A) which are most vital between 2020 and 2040 because they have bad prognosis for the end of the century would mean defying silvicultural reality. To promote large-scale forest conversion with alternative (sub)Mediterranean species (C) that are still little known is also not realistic, but forestry is strongly advised to gain experience with these species today [99]. The key of a climate adaptive forestry is therefore species mixture—a general demand in any forest climate change literature [15,102–105]. However,

it is not just any mixture but a mixture that considers elements of each of the three groups A, B, C. It is a mixture that neither establishes nor sustains itself. As pointed out, the important group (B) can only survive in mixture with (A) today if it is actively promoted. The alternative species (C) must be actively migrated. Optimized planting schemes with respect to the difficult to obtain and expensive alternative species are published in [106].

Of course, the vicinity of species on a climate scale or a geographic scale is distinct from the temporal scale. In other words, whereas analogues (and predictions from species distribution models do not differ in this respect) can substitute space for time the forests cannot. Species change will not be gradual and not happen all by itself. The "legacy of the established" is very strong due to forest-inherent resilience and forestry tradition [96,97,107]. Shifts in the tree ranges at the cold edge are too slow [43,46,108,109], to keep pace with the expected shifts in climate [44,110] and the gap is widening. Consequence is a climatic debt [111], extinction debt [112] or resilience debt [107] which is bound to be "catalyzed by disturbance" [6]. To work towards healthy forest and forest functions in a changing climate requires an actively assisted shift in the species spectrum [45,47,48,51,52].

**Supplementary Materials:** The following are available online at https://www.mdpi.com/article/10.3390/su13126522/s1, Figure S1: Climate distance maps of Europe for Roth RCP 8.5 mean variant: (a) 2000, (b) 2060, (c) 2100, Figure S2: Prevalence trajectory graphics showing relative prevalence from 2000 to 2100 of 23 major tree species in the twin regions of the RCP 4.5 and RCP 8.5 low and high variants for site Roth. Grey numbers below the x-axis indicate the number of NFI-plots in the twin regions of each time step; grey numbers on the right vertical axis the total number of species occurrence in the plots of all twin regions from 2000 to 2100. Asterisks <*> in the species cones mark the three species with the highest absolute prevalence in each 20-year time step. Species abbreviations as in Table 3 (main article). Table S1: Model weights assigned to each climate model to calculate the mean, low and high variant for RCP 4.5 and RCP 8.5 (for model references c.f. Table 1, main article), Table S2: NFI-plots = 1 km$^2$ grid cells with one or more NFI-(sub)plots [59]; forest area from *State of Europe's forests 2015*; plot representation area = average forest area per NFI plot.

**Author Contributions:** Conceptualization, T.M., C.K.; methodology, T.M., S.B., C.K.; software, T.M., S.B.; validation, T.M., S.B., C.K.; formal analysis, T.M., S.B., C.K.; investigation, T.M., S.B., C.K.; resources, T.M., S.B., C.K.; data curation, T.M.; writing—original draft preparation, T.M.; writing— review and editing, T.M., S.B., C.K.; visualization, T.M., S.B.; supervision, C.K.; project administration, T.M., C.K.; funding acquisition, T.M., C.K. All authors have read and agreed to the published version of the manuscript.

**Funding:** This research was funded by the German forest climate funds of the federal ministry of food and agriculture and the federal ministry for the environment, nature conservation and nuclear safety on behalf of a decision of the German Bundestag, grant number 22WK514405. The APC was funded by Bavarian state institute of forestry.

**Institutional Review Board Statement:** Not applicable.

**Informed Consent Statement:** Not applicable.

**Data Availability Statement:** Not applicable.

**Conflicts of Interest:** The authors declare no conflict of interest. The funders had no role in the design of the study; in the collection, analyses, or interpretation of data; in the writing of the manuscript; or in the decision to publish the results.

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
