# Peer review of "Climate Analogues for Temperate European Forests to Raise Silvicultural Evidence Using Twin Regions"

_sustainability, doi:10.3390/su13126522_

Round 1

Reviewer 1 Report

Title: Climate analogues for temperate European forests –forestry practice profits from silvicultural evidence in twin regions

Abstract: This section looks good as it captures the study

Introduction: This section is also well well written

Materials and Methods: This section is clear and well described. I will only suggest site photos and some descriptions are provided in the beginning.

The Discussion section is unclear and distorted. I suggest a complete revision of the discussion section (should properly compared with similar studies) and a concise conclusion provided.

The paper makes interesting reading and it is an important addition to literature. However, I suggest the discussion section is redone and conclusion added. Also English grammar should be checked.

Reviewer 2 Report

After a careful review of the manuscript submitted, some of the main issues are indicated below:

The objectives through the entire manuscript are not firmly stated (e.g. Abstract) and the approach that does not fulfil the objectives of the work.

Moderate and intense climate change scenarios (RCP 4.5 and RCP 8.5) do not cover the uncertainty on future climate. It should be clearly explained the selection of the 2 scenarios, since all the RCP scenarios are expected to have significant effects on ecosystems structure and functioning.

Even though analogues approach is correct in many situations, it has strong disadvantages when applied to ecosystems. Analogues do not take into account forest resistance and resilience and processes like species seed dispersal, species recruitment, competition... In addition, even when a given region is climate analogue of a region of interest for future climate, the future trajectory of our ecosystems does not necessarily go through such structure and functioning because the potential analogue ecosystem nowadays might be vulnerable under current climate.

The introduction section is not a proper state of the art and justification, and includes information that should be relocated to Material and Methods and to Results. The different sections include redundant and sometimes unneeded information and sometimes they don't fully cover the expectation instead, like a proper state of the art.

The information in the manuscript is not fully consistent/coherent. Some paragraphs should either to include citations to justify the information or to clearly state that they are assumptions in this work. Some concepts/terms (e.g. hotspots) are not correctly used; it is essential to use them properly or their meaning stops being accurate.

Writing is not fluent and needs to be rewritten; it complicates the communication of ideas and the understanding. Some information in the main text of the manuscript should not be there, but included in figure footnotes (e.g. line 290... ). Some articles unneeded; some verbal tenses are incorrect and some adjectives, adverbs and nouns are improperly located or incorrectly used (e.g. "To quantitative") and the text should be carefully reviewed. It is common that a word/term is repeatedly used within the same paragraph. Some expressions, e.g. "like pearls", should not be used in scientific publications.

Forest ecosystems applying this approach we are disregarding the ecosystems inertia.

This approach also disregards ecosystems history.

can already be in suboptimal conditions

Reviewer 3 Report

General comments

Selecting a forest site based on the model assumption that this site vegetation reflects vegetation succession expected under a future climate change (or warming) scenario, and then comparing its vegetation with existing forest vegetation is a valuable theoretical idea and merits appreciation. However, it seems that authors have assumingly established that forestry readers would know their ideas or concepts and they do not need to put up a case in a sentence or two in the abstract. They just started narrating things based on their assumptions. Clearly, authors have good English language skills but their write-up is full of implicitness, specifically in title and abstract. Let me go ahead and see how the later sections are written – hopefully, they are explicitly based on strong assumptions or facts.

The title is unclear maybe due to authors assumed that readers already know their experiment; the reader could feel hard to connect the two parts of the title that how climate analogues would relate to forestry practice and silvicultural evidence. No clue but a wild guess is the option a reader could make. The title needs to be interesting and inviting to further read the article. Likewise, the abstract seems to be consistently assuming that forestry readers would know what RCP 4.5 and 8.5 models are? – a technically unprofessional write-up of an abstract.  The abstract seems to say a lot before presenting providing with a single sentence result/finding.  

Specific comments

Line 33: …… consequence of. Do authors mean climate extreme a warming or warming + drought and/or altered precipitation?

Line 107-112: It would be great to shift or fit some or abridge of this paragraph to/in the introduction section to help readers understand the climate analogues beforehand

Line 114: 2 meters from where?

Line 114-115: not sure how authors would include climate extremes (earlier mentioned) when the resolution is that coarse (monthly average). A reference may support the method. Do RCP 4.5 and 85 are based on monthly averages?

Line 123-124: what is statistical downscaling here?

Line 124-125: any adiabatic processes included?

Line 144-146: Now authors are taking seasonal means compared to the monthly means and random events

Fig. 1C. Do authors find some relationships between seasonal precipitation (actually rainfall) and temperature?

Fig. 5, 6: are all same degree curves?.

The variance may need to be a little darker as they may not be visible in case of printing

 I like the later part of the discussion where a reader has a clear take-home message which is clearly in alignment with the set of objectives and in relation to hypotheses and/or assumptions made in the intro section.

Reviewer 4 Report

The paper is dedicated to a very important, but still less studied topic of climate change impact on forestry at macro-scale. As forests are a complex ecosystem with multiple flows of services, their reaction on change of external climatic conditions may be described only with quite a high uncertainty. Thus, every piece of knowledge on that topic is of a great importance.

Authors made a successful attempt to compare “twin” climatic regions and calculate “climatic distances” between future climate trajectories and establish some suggestions on how to manage tree species composition, silvicultural practices and pest management aiming at mitigation of climate change negative effects in the future. 

Overall impression on this paper is favorable.

However, some minor things:

  • spaces around dash between “forests” and “forestry” in the title are not even.
  • L 203. “To quantitative” — what does that mean? To quantify? A comma also seems to be missing after “are” in this sentence.
  • Eq (1). What does the “~” sign means in this context? If it is not a strict equation than the meaning of this mathematical (logical) relation might be explicitly explained in the text. Otherwise, it is not obvious, how you operationalize this formulae in your future calculations.
  • What don't you use common mathematical symbols for sum and sqrt in your equations? It seems that the introduction of letter designation of this operators is not proven.

Round 2

Reviewer 1 Report

The Authors have addressed all the concerns I raised as well as those raised by the other reviewers.
I appreciate the Authors effort in revising the manuscript.

Author Response

Dear reviewer,

Thanks for your positive reply! Further changes have been made in this second revision in response to an editor's review. The former references to climate analogues in the introduction have been extended into a concise state-of-the-art summary. A more detailed site description has been added.

Sincerely, Tobias Mette (also in the name of co-authors)